# The Therapeutic Potential of Agarwood as an Antimicrobial and Anti-Inflammatory Agent: A Scoping Review

**DOI:** 10.3390/antibiotics13111074

**Published:** 2024-11-12

**Authors:** Aswir Abd Rashed, Mohd Azerulazree Jamilan, Salina Abdul Rahman, Fatimah Diana Amin Nordin, Mohd Naeem Mohd Nawi

**Affiliations:** Nutrition, Metabolism and Cardiovascular Research Centre (NMCRC), Institute for Medical Research, National Institutes of Health, Ministry of Health, Persiaran Setia Murni U13/52 Setia Alam, Shah Alam 40170, Selangor, Malaysia; azerulazree@moh.gov.my (M.A.J.); sar@moh.gov.my (S.A.R.); fatimahdiana@moh.gov.my (F.D.A.N.); naeem@moh.gov.my (M.N.M.N.)

**Keywords:** *Aquilaria* spp., antimicrobial, anti-inflammatory

## Abstract

Background/Objectives: Microorganisms such as bacteria, viruses, and fungi are frequently the cause of infections. Antimicrobial agents, such as antibiotics, antivirals, and antifungals, are used to target and eliminate these infectious agents. On the other hand, inflammation is a natural response of the immune system to injury, infection, or irritation. Although herbal remedies have been used to treat these conditions for centuries and can be effective in certain situations, it is crucial to use them with caution. Not all herbal remedies are supported by scientific evidence, and their safety and efficacy can vary. Thus, we conducted this review to determine the potential health benefits of agarwood as an antimicrobial and anti-inflammatory agent. Methods: Three databases (PubMed, Scopus, and Google Scholar) were used to search for original papers submitted between 2013 and 2023, using the Medical Subject Heading (MeSH) terms “agar-wood” crossed with the terms “antimicrobial” and/or “anti-inflammatory”. Synonyms and relevant search terms were also searched. Results: The most-studied agarwood for antimicrobial and anti-inflammatory agents is *Aquilaria sinensis*. Some studies have shown its potential application as a potent inhibitor of fungi, including *Lasiodiplodia theobromae*, *Fusarium oxysporum*, and *Candida albicans*. Moreover, it is capable of inhibiting *Bacillus subtilis* and *Staphylococcus aureus* activities. Several chromones detected in agarwood have been shown to inhibit NF-κB activation, LPS-induced NO production, and superoxide anion generation. In conclusion, more research is needed, particularly regarding future intervention studies, to enhance our knowledge and understanding of agarwood and its isolates. Conclusions: This review reveals that despite the absence of clinical trials, agarwood exhibits antimicrobial and anti-inflammatory properties.

## 1. Introduction

The primary source of agarwood, also known as oud, is Aquilaria trees. These trees are native to several Asian nations, mainly located in Southeast Asia. In some countries, Aquilaria trees grow in natural forests and in plantations [1]. In countries such as Thailand, Cambodia, Vietnam, Malaysia, Indonesia, Laos, India, Bangladesh, and parts of China, Aquilaria trees that produce agarwood are often found in natural forests [2]. Specific locations where agarwood is found within these countries can vary due to climate and soil conditions [3]. Certain ecological conditions, such as hilly or mountainous regions with particular soil types and sufficient rainfall, are more conducive to the growth of these plants. In recent years, some countries have tried cultivating Aquilaria trees in plantations. These plantations aim to sustainably produce agarwood by stimulating the resinous formation in the trees. Plantations can be found in Southeast Asian countries such as Malaysia and Indonesia [4].

While some preliminary research has suggested the presence of antimicrobial properties in agarwood, concrete scientific evidence supporting this claim is limited, and most studies were in the early research stages. It has been demonstrated that agarwood leaf ethanol extract exhibits antibacterial action against both bacteria and fungi. The extract contains bioactive compounds like flavonoids and tannins which contribute to this antimicrobial activity [5]. Agarwood produces oud oil, which is rich in a variety of volatile compounds that include sesquiterpenes such as α- and β-guaiene, agarospirol, agarol, and various other sesquiterpenoids and phenylethyl chromones [6]. These bioactive substances found in agarwood have been shown to possess antimicrobial properties. The general potential mechanisms through which agarwood might exert antimicrobial effects could include (a) the disruption of cell membranes, by which agarwood’s compounds have the potential to cause the microorganisms’ cell membranes to rupture, allowing internal organelles to seep out, eventually causing cell death [7]; (b) interference with microbial enzymes, whereby the components of agarwood may block or obstruct vital microbial enzymes, impairing the metabolic activities that are necessary for microbial viability [8]; (c) antioxidant effects that may help fight microbial infections by modifying microbial viability and lowering oxidative stress [9]; (d) the modulation of gene expression by compounds found in agarwood that may affect the pathogenicity, survival, or replication of the microbes [10]; (e) impact on the biofilm formation of bacterial colonies that are frequently resistant to antimicrobial treatments [11].

Additionally, agarwood’s potential anti-inflammatory properties have also been investigated. Agarwood extracts have shown potential in inhibiting various inflammatory mediators such as cytokines, prostaglandins, and leukotrienes [12]. These mediators play critical roles in the inflammatory response, and the ability of agarwood to inhibit their release or activity suggests anti-inflammatory potential. Some studies indicate that agarwood extracts may inhibit enzymes like cyclooxygenase (COX) and lipoxygenase (LOX) involved in the generation of inflammatory mediators including prostaglandins and leukotrienes [7]. By inhibiting these enzymes, agarwood could potentially suppress the inflammatory cascade. Moreover, agarwood may modulate the immune response, potentially influencing the activities of immune cells involved in inflammation. Its immunomodulatory effects could help regulate inflammatory responses. Additionally, agarwood might interfere with specific cellular signaling pathways involved in inflammation, such as NF-κB (nuclear factor kappa-light-chain-enhancer of activated B cells) and MAPKs (mitogen-activated protein kinases), which are associated with the regulation of inflammatory responses. Inflammation often involves oxidative stress, and the antioxidant activity of agarwood may help neutralize free radicals and reduce inflammation induced by oxidative stress. Thus, this scoping review was conducted to determine the effect of agarwood as a potential antimicrobial and anti-inflammatory agent.

## 2. Results

We examined 321 publications in the initial stage before screening. Following screening, 282 out of 313 articles were excluded according to established inclusion and exclusion criteria. Articles that have met the requirements of being written in English or Malay, having available full text, and being peer-reviewed were included, ensuring the selection of high-quality and relevant research. In contrast, review papers, letters to the editor, or duplicate articles were excluded, as these did not contribute original research data or were redundant (Figure 1). We also limited the publication period to 10 years for the following reasons:(a)Relevant and recent publications are required to ensure that the review reflects the most current research and knowledge.(b)Limiting the timeframe helps to maintain the focus on recent developments and trends within a manageable scope.(c)Limiting the publication period makes the reviewing process more feasible in terms of time and effort.

A more refined search, taking into account the availability of peer-reviewed publications, full text papers, and library collection access, yielded a total of 29 articles. Only 27 full-text publications were deemed relevant after additional evaluation, and these were included for final review (Table 1). For a more thorough evaluation of the evidence supporting the efficacy of agarwood as a possible antibacterial and anti-inflammatory agent, all relevant publications were printed out. While the ten-year limit is standard, it can vary depending on the field, topic, and specific requirements of the review paper. Ultimately, the timeframe chosen should balance comprehensiveness with relevance to provide the most insightful and valuable literature synthesis.

## 3. Discussion

Most studies on agarwood have been related to its aromatic properties, its traditional applications in perfumes, and its potential medicinal benefits when used topically or inhaled through aromatherapy. While agarwood is not commonly consumed as a food or part of a regular diet, it has been utilized in some cultures for its purported medicinal properties. In certain traditional practices, agarwood has been used in minimal quantities or as an ingredient in herbal remedies for the treatment of various health conditions such as digestive issues, asthma, pain relief, and it has even been used as an aphrodisiac [7]. Agarwood may also be found in traditional medicines such as tea or extracts. As with any herbal or natural remedy, it is essential to be aware of potential allergic reactions, interactions with medications, and the lack of scientific evidence supporting its effectiveness and safety for consumption. Therefore, the safety and efficacy of consuming agarwood for these purposes must be extensively studied in clinical trials. It is necessary to highlight that the consumption of agarwood or its derivatives should be approached with caution and under the guidance of a healthcare professional due to limited scientific evidence regarding its safety and potential side effects. As of our knowledge cutoff date of January 2023, more scientific research explicitly focusing on agarwood consumption was still needed.

At the initial stage, we tried to separate antimicrobial and anti-inflammatory activities. However, during the write-up process, we found that combining antimicrobial and anti-inflammatory activities offers several advantages and synergistic effects. Thus, it is suitable to incorporate them because many infections are accompanied by inflammation, and vice versa. Treatment can provide a more comprehensive therapeutic approach by targeting antimicrobial activities to combat the disease and anti-inflammatory activity to reduce inflammation and its associated symptoms. Moreover, research regarding compounds or formulations that exhibit antimicrobial and anti-inflammatory properties can develop novel therapeutic agents with unique mechanisms of action and improved therapeutic profiles. Overall, combining antimicrobial and anti-inflammatory activities represents a promising medical and pharmaceutical research approach, aiming to provide more effective, targeted, and holistic treatments for infectious and inflammatory diseases.

In this scoping review, we found that a group of researchers had investigated the chemical composition of agarwood from different Asian countries, and they discovered that the main volatile components were oleic acid 3-(octadecyloxy) propyl ester, 3-ethyl-5-(2-ethylbutyl)-octadecane, and docosanoic acid 1,2,3-propanetriyl ester [13]. Interestingly, the main active ingredients of *A. sinensis* were aromatic compounds, sesquiterpenoids, and chromone compounds. They also recorded that agarwood displayed more significant antibacterial effects against Gram-positive than against Gram-negative bacteria. This effect may be due to the LPS layer on the Gram-negative bacteria cell wall, which prevents hydrophobic compounds from entering the cells and reduces the bacteriostatic impact [40]. The inhibition rates they obtained were in the following order: *S. aureus* > *B. subtilis* > *E. coli* [13].

Canli et al. conducted the first study to screen the antimicrobial properties of the ethanolic extraction of *A. agallocha* roots in vitro [14]. *A. agallocha* is considered to be synonymous with *A. malaccensis*. By performing a disk diffusion method on 17 bacteria and one fungus (Escherichia, *Bacillus*, *Enterococcus*, *Salmonella*, *Candida*, *Enterobacter*, *Klebsiella*, Staphylococcus genera, *Listeria*, and *Pseudomonas*), they found that ethanol extracts were active against most of the strains, especially against *E. faecium*, *L. monocytogenes* ATCC 7644, *B. subtilis* DSMZ 1971, *C. albicans* DSMZ 1386, *S. epidermidis* DSMZ 20044, and *S. aureus* ATCC 25923. Unfortunately, they did not identify the active substances elucidating the mechanism of action involved in this antimicrobial activity.

Another group of researchers also reported the antimicrobial effect of agarwood on *Proteus mirabilis* and *S. aureus* [15]. Using the Kirby–Bauer disc diffusion assay, they discovered that the average diameter zones blocked by the ethanol extract of agarwood leaves in *S. aureus* measured 12.50 mm (300 mg/mL), 13.51 mm (400 mg/mL), and 15.80 mm (500 mg/mL). The study also reported that the average diameter zone blocked by the ethanol extract of agarwood leaves in *P. mirabilis* concentrations of 300 mg/mL, 400 mg/mL, and 500 mg/mL were 12.10 mm, 13.26 mm, and 15.19 mm, respectively. As previously reported by Wang et al. [13], this study also confirmed that ethanol extracts of agarwood leaves have antimicrobial properties against selected Gram-positive and Gram-negative bacteria.

In another study, an ethanol extract of *A. malaccensis* leaf was also tested for its antimicrobial activity against selected fungi and bacteria that grow on the skin [16]. It was found that the antibacterial activity of the extract (1.25–20% concentration) showed an auspicious result against *S. aureus*, where it was categorized as susceptible at a 5% concentration. However, *S. epidermis* and *Propionibacterium acnes* at 20% concentrationd are categorized as intermediate, whereas all other tested concentrations are categorized as resistant. The study also reported the anti-fungal activity of *C. albicans*, which was classified as intermediate at 20%, whereas the other tested concentrations were classified as resistant; however, other fungi of *Trichophyton* sp. showed the inhibitory zone as resistant for all concentrations. This suggests that flavonoids, tannins, and triterpenoids are the active compound groups that contributed to its antimicrobial activity. The antimicrobial efficacy of *A. crassna* leaf aqueous extract against *S. epidermidis* has also been found [17]. The extract suppressed *S. epidermidis* growth in 2 mg (12.0 ±1.0 mm), 4 mg (15.0 ± 0.4 mm), and 6 mg (18.0 ± 1.0 mm) concentrations, as determined by disc diffusion assay. *S. epidermidis* was disposed by the extract with the MIC and MBC of 6 and 12 mg/mL, respectively. This action is due to the disruption of the biofilm and ruptured cell walls, which ultimately changed the shape of the bacteria.

Sometimes, an ingredient has a better effect when mixed with other active ingredients. Jihadi et al. [18] reported the first work performed using a combination of polymyxin B and *A. malacensis* extract via an in vitro study targeting *Acinetobacter baumannii* and *Klebsiella pneumonia*. Polymyxin B is an antibiotic that belongs to the polymyxin group of antibiotics. In this study, polymyxin B was utilized at a clinically relevant dose of 1 μg/mL, with susceptibility breakpoints at MIC of ≤2 μg/mL [41]. The performance of this combination extract was assessed using in vitro time-kill tests and analysis of GC-MS at 4 and 24 h. The researchers discovered that crude extract, either combined with polymyxin B or used alone, significantly decreased and inhibited bacteria over a 24 h period. However, the combination yields a considerably higher bactericidal effect exceeding ≥3 log10 CFU/mL at the end of the 24 h research period, notably for the extract at 64 mg/mL, compared to the results for polymyxin B. alone, which were only about ≥1 log10 CFU/mL. The GC-MS investigation of *A. malaccensis* ethanolic crude leaf extract revealed over sixty compounds, including a large amount of phytol and 9,12-octadecadienal. Compounds believed to contribute to the extract’s antimicrobial activity include phytol, 9,12-octadecadienal, oleic acid, n-decanoic acid, n-hexadecanoic acid, and squalene.

In another intriguing investigation, *A. malaccensis* was employed as a biogenic medium to produce CuO NPs with antimicrobial properties [19]. The boiled leaf extract reacted with 5 mM CuSO_4_.5H_2_O at pH 6 and was incubated at 70 °C without shaking, resulting in a high rate of CuO NPs production and exhibiting a UV absorbance peak of 430 nm. Field emission scanning electron microscopy (FESEM) and transmission electron microscopy (TEM) indicated that the nanoparticles are primarily spherical, with sizes ranging from 6 to 32 nm. Antimicrobial investigations revealed that 20 μL and 40 μL of 70 μg/μL CuO NPs effectively inhibited the Gram-positive bacteria *B. subtilis*, with average zones of inhibition measuring 24.43 ± 0.10 mm and 27.31 ± 0.13 mm, respectively.

In Vietnam, effects of essential oils derived from the trunk of Vietnam-originated agarwood of *A. banaensis P.H.Hô* against *B. subtilis, S. aureus*, *E. coli*, *L. fermentum*, *P. aeruginosa*, *S. enteric*, and *C. albicans* were explored. The researchers discovered that the essential oil from the trunk had a more promising antimicrobial impact than the leaf extract [20]. The study examined the chemical composition of the leaf extract, which included β-selinene, β-caryophyllene, β-elemene, α-humulene, α-selinene, and β-guaiol. In contrast, the trunk contained oleic acid, tetradecanoic acid, and hexadecanoic acid, which could explain the discrepancies in the results for these two extracts as possible antimicrobial agents.

During the COVID-19 pandemic, agarwood was also explored for its effectiveness using an in silico approach. The potential antiviral activity of oleanin triperpenoids in agarwood against coronavirus 2 (SARS-CoV-2) has been investigated using Lipinski’s rule of five and the prediction of absorption, distribution, metabolism, and excretion (ADME) [21]. The study found that four oleanin triperpenoids, 11-oxo-β-amyrin (ΔG = −9.8 kcal/mol), hederagenin-an (ΔG = −9.6 kcal/mol), 3β-acetoxyfriedelane (ΔG = −9.4 kcal/mol), and ursolic acid (ΔG = −9.5 kcal/mol), displayed a higher affinity to coronavirus 2 (SARS-CoV-2) than did lopinavir (ΔG = −6.2 kcal/mol) and remdesivir (ΔG = −7.2 kcal/mol) when molecularly docked in the main protease (Mpro) receptor. The prediction of ADME contributed to the development of hederagenin, a potential oral medication in which several primary amino acids, including methionine 49 and 165, glutamine 189, proline 168, threonine 25, and arginine 188, were engaged in the interactions.

Based on the study of the phytochemical properties of distilled water and various parts of the agarwood plant (*A. malaccensis* Lamk), the secondary metabolites of the glycosides were found to be responsible for the antimicrobial activity derived from several agarwood parts, including the leaf, trunk, skinned stem, and bark [22]. The distilled water from *A. malaccensis* Lamk exhibits antimicrobial activity against *Streptococcus* mutants and is effective in resisting the bacteria.

Conversely, Dahham et al. [23] isolated β-caryophyllene from *A. crassna* essential oil, a chemical classified as a terpene. β-caryophyllene was identified as the largest component at 8.1%, followed by 1-phenanthrenecarboxylic acid at 7.1% and 2-naphthalene-methanol at 6.2%. The antimicrobial action of β-caryophyllene was evaluated against human pathogenic bacteria and specific strains of fungi. β-caryophyllene showed substantial antibacterial effect against all examined microbial strains, including *B. subtilis*, *S. aureus*, *B. cereus*, *P. aeruginosa*, *A. niger*, *E. coli*, *P. citrinum*, *R. oryzae*, *K. pneumoniae*, and *T. reesei*, as well as significant anti-fungal activity compared to that of kanamycin.

In recent years, utilizing plant extracts as biopesticides for nanoparticle synthesis has gained popularity due to its low cost, eco-friendliness, and single biosynthesis approach. In Indonesia, Prasetya [24] explored the antibacterial activity of *A. agallocha* oil nanoemulsion against multidrug-resistant bacteria (MDR) and antibiotic-nonresistant bacteria inlcuding *S. aureus* (ATCC 43300), *E. coli* (ATCC 35218), *E. coli* (ATCC 25922), *K. pneumoniae* (ATCC 700603), *K. pneuomoniae* (ATCC 8724. T), and *S. aureus* (ATCC 25923). The study reported that with a 99.35% transmittance, the 1% agarwood oil nanoemulsion concentration exhibited the lowest size, measuring 17.7 nm. The inhibition zones for *S. aureus* ATCC 25923 and *K. pneumoniae* ATCC 8724 were 2.6 mm and 3.3 mm, respectively. The non-resistant *E. coli* ATCC 25922 showed an inhibitory zone of 13.3 mm. Based on their findings, the researchers suggested that a higher concentration of the oil nanoemulsion may be required to boost its inhibitory action against MDR than that required for non-resistant antibiotic bacteria. In another meticulous study on nanoparticles, Ga’al et al. [25] thoroughly investigated *Pogostemon cablin* essential oil (PcEO) and biosynthesized silver nanoparticles (AgNPs) against the larvae and pupae of the dengue and zika virus vector Aedes (Ae) albopictus. Their research, which included formation and biophysical characterization, was conducted with utmost care and attention to detail. The study found that biofabricated AgNPs were the most hazardous to Ae. Albopictus when compared to the results for the evaluated essential oils. The LC50 values of AsEO ranged from 44.23 (I) to 166 (pupae), PcEO ranged from 32.49 (I) to 90.05 (IV), AsEO-AgNPs ranged from 0.81 (I) to 1.12 (IV), and PcEO-AgPNs ranged from 0.85 (I) to 1.19 (IV). Their research also revealed that the synthetic AgNPs had a significantly greater impact on the epithelial cells and brush borders of both the control and treated larvae when compared to that of the essential oils (AsEO and PcEO).

Meanwhile, Chen et al. [26] reported that the chemical method improved the quality of the agarwood derived from *A. sinensis* (S1) compared with that obtained from wild agarwood (S2) and healthy trees (S3). They also evaluated the antimicrobial activities of that particular essential oil. From GC-MS chromatograms, they determined the similarity of essential oils of S1 and S2, with a high concentration of sesquiterpenes and aromatics. However, S3’s essential oil included a significant concentration of fatty acids and alkanes. The essential oils of S1 and S2 were more effective at inhibiting *S. aureus* and *B. subtilis*. Even at the maximum experimental dose of 2 mg/mL, the three extracts showed little activity against *E. coli*.

*Lasiodiplodia theobromae* (F) was utilized in a work by another researcher to stimulate *A. sinensis* (Lour.) Gilg to produce agarwood [27]. The chemical composition of F was determined using GC-MS. Their findings revealed that essential oils derived from *A. sinensis* generated by *L. theobromae* were chemically and antimicrobial similar to those derived from wild agarwood (W). F’s essential oil was identical to W’s, with a high concentration of sesquiterpenes and aromatic components. However, the essential oil of uninoculated healthy trees (H) contained a high concentration of alkanes. Regarding their anti-fungal activity, F’s and W’s essential oils were more potent inhibitors of *F. oxysporum*, *L. theobromae*, and *C. albicans* than were H’s essential oil.

In the same family as *Aquilaria*, *G. versteegii* fruit extract was assessed for antimicrobial activity against *S. aureus* and *E. coli* by Hidayati et al. [28]. *G. versteegii* is a tree from the *Thymelaeaceae* family, known to produce agarwood. The researchers used n-hexane, dichloromethane, and methanol for the extraction process and employed agar well diffusion and GC-MS methods for analysis. The dichloromethane extract, particularly at a 40% concentration, exhibited the most potent antimicrobial activity, with a 13.17 mm inhibition zone against *S. aureus*, compared to 7 mm for *E. coli*. The extracts demonstrated total and partial inhibition against *S. aureus* and *E. coli*. Several compounds were identified by GC-MS analysis, including oleic, stearic acid, palmitic, 2,3-dihydro-3,5-dihydroxy-6-methyl-4H-pyran-4-one, squalene, bis-(2-ethylhexyl) phthalate, methyl octadec-9-enoate, and 2-monopalmitin derivatives. The study revealed a more potent antimicrobial effect against *S. aureus*, particularly with the 40% concentration of dichloromethane extract, and identified various compounds in the *G. versteegii* fruit extracts.

For studies that assess agarwood as an anti-inflammatory agent, Yu et al. [29] investigated the anti-inflammatory properties of 5,6,7,8-tetrahydro-2-(2-phenylethyl) chromones derived from *A. sinensis* by utilizing the RAW 264.7 cell inhibition effect on lipopolysaccharide (LPS)-induced nitric oxide (NO) release. Among the novel compounds, Compound 2 [(5S,6R,7S,8S)-8-chloro-5,6,7-trihydroxy-2-(2-phenylethyl)-5,6,7,8-tetrahydrochromone] displayed the highest IC50 value of 3.46 μM, indicating considerable anti-inflammatory action.

Like many other terpenes and natural compounds, sesquiterpenes have sparked widespread interest due to their involvement in biological systems and their potential medicinal use. In 2019, Yu et al. [30] recorded the existence of eleven sesquiterpenes in *A. sinensis*. Compound baimuxinol (1) was among the new natural products found. They recorded that compounds 1,4 and 9 displayed anti-inflammatory activity, with IC50 values of (2.5 ± 0. 35), (3.2 ± 0.2), and (4.3 ± 0.56) μmol/L, respectively.

Flavonoids are abundant in medicinal plants and fruits and exhibit a variety of pharmacological actions. A flavonoid compound, pilloin, isolated from *A. sinensis*, was tested for anti-inflammatory action in vitro and in vivo [31]. This compound’s ability to downregulate pro-inflammatory cytokines (e.g., TNF-α and IL-6), as well as enzymes (e.g., iNOS and COX-2), to inhibit NF-κB and MAPK signaling pathways in LPS-activated macrophages, and to suppress the phenotypes and functions of activated macrophages (i.e., ROS production and phagocytic activity) holds great promise for clinical applications. Pilloin’s reduction of LPS-induced cytokine production (e.g., TNF-α and IL-6) in the serum and tissues of septic mice could potentially lead to novel therapeutic strategies.

Wang et al. [33] conducted an extensive study on the anti-inflammatory properties of several *A. sinensis* compounds. The methanolic extract from resinous *A. sinensis* has shown remarkable potential in inhibiting NF-κB activation in RAW 264 LPS-stimulated macrophages. The relative luciferase activity values ranged from 0.31 ± 0.05 to 0.55 ± 0.09, which were significantly lower than the vehicle control value of 1.03 ± 0.02. This finding indicates a compelling effectiveness that merits further exploration. Moreover, some compounds might decrease LPS-induced NO generation in RAW 264.7 cells, without cytotoxicity, after 24 h of treatment [33]. In addition, several chromone-related compounds exhibited more than an 80% inhibition towards superoxide anion production by human neutrophils at 50 μM of formyl-l-methionyl-l-leucyl-l-phenylalanine (fMLP) [35]. Moreover, the methanolic extract of several compounds of *A. sinensis*’ stem bark inhibited (IC50 ≤ 12.51 μM) superoxide anion production in human neutrophils’ response to fMLP/cytochalasin B. Additionally, 7-Hydroxy-6-methoxy-2-(2-phenylethyl)chromone, velutin, 3′-hydroxygenkwanin, 6,7-di-methoxy-2-(2-phenylethyl)chromone, and ergosta-4,6,8(14),22-tetraen-3-one demonstrated the most remarkable effectiveness among the isolates (IC50 values ≤ 15.25 μM) by inhibiting elastase release induced by fMLP/CB [36].

Anti-inflammatory effects of 2-(2-phenylethyl) chromone derivatives using ethanolic extract of resinous *A. sinensis* have also been reported by Huo et al. [37]. Their preliminary analysis of the structure–activity relationship documented that the presence of a chlorine substituent and an epoxy group on the A-ring were associated with the anti-inflammatory activity of 5,6,7,8-tetrahydro-2-(2-phenylethyl)-chromones. Compounds 2–4, 11, 12, and 15 significantly inhibited NO generation in LPS-stimulated RAW 264.7 cells, with IC50 values of 1.6–7.3 μM. Additionally, these compounds showed no significant cytotoxicity (up to 100 μM) after 24 h of LPS treatment, as measured using the MTT technique. In another study utilizing the same ethanolic extraction method, several compounds displayed a considerable reduction of NO generation in LPS-stimulated RAW 264.7 cells, with IC_50_ values ranging from 7.0 to 12.0 μM and no cytotoxicity effect (up to 80 μM) after 24 h of LPS treatment [34]. Huo et al. [32] used LC–MS-guided fractionation to isolate fifteen previously undescribed 2-(2-phenylmethyl)chromone dimers from *A. sinensis* and two known analogues. As might be expected, these isolated compounds effectively inhibited NO generation in LPS-stimulated RAW 264.7 cells, with IC_50_ values ranging from 0.6–37.1 μM, indicating their potential as anti-inflammatory agents. These findings highlight the possible anti-inflammatory capabilities of these chromone derivatives, emphasizing their therapeutic potential, while not inducing cytotoxic effects at relevant concentrations.

*A. sinensis* leaves have previously been found to exhibit analgesic and anti-inflammatory properties [42]. *A. sinensis* leaves also contain eight α-glucosidase inhibitors [43], which may be utilized as a traditional treatment for diabetes. However, based on an in vivo study by Sattayasai et al. [38] using the rat paw assay at 800 mg/kg, methanolic *A. crassna* leaf extract showed no anti-inflammatory activity. Nevertheless, the extract possesses antipyretic, analgesic, and anti-oxidative properties.

The results of a study published by Peng et al. [39] investigating incense smoke from agarwood revealed low amounts of TNF-α and IL-1α derived from normal inactivated RAW 264.7 cells; however, with LPS exposure, higher amounts were obtained after 24 h of incubation. Indomethacin treatment resulted in concentration-dependent AAW, BCDA, and AWIT decreases at 20, 40, and 80 μg/mL TNF-α, and the IL-1α levels were substantially more significant than those in the normal group (*p* < 0.05 or *p* < 0.01), indicating improved anti-inflammatory effects.

## 4. Materials and Methods

Three databases (PubMed, Scopus, and Google Scholar) were used to search for original papers published from 2013 to 2023 using the Medical Subject Heading (MeSH) terms “agarwood” crossed with the terms “antimicrobial” and/or “anti-inflammatory”. Publications with available full papers were assessed, and only studies published in English and Malay were considered for evaluation. Papers including human and clinical studies of agarwood were included. Letters to the editor and reviews, however, were not included. Duplicate articles were eliminated.

## 5. Limitations of the Study

This review paper has several limitations because some information cannot be fully obtained from the original article. The use of various tested materials, such as the type of solvent used in the extraction of either ethanol or methanol, the type of essential oil, and the type of nanoemulsion, also makes it difficult to draw more accurate conclusions. The small number of studies also limits the ability to identify the effectiveness of agarwood.

## 6. Conclusions

We examined in depth twenty-seven relevant studies worldwide related to agarwood as a potential anti-inflammatory and antimicrobial, consisting of experiments involving chemical composition, in silico, in vitro, in vivo, and combined in vitro and in vivo methods. We found that the active compounds in agarwood positively affect its effectiveness. More research is needed, particularly regarding future intervention studies, to enhance our knowledge and understanding of agarwood and its isolates. These promising compounds are ripe for biomedical exploration and could become powerful tools in treating and preventing microbial and inflammatory diseases. Investing in this research could lead to groundbreaking advancements in healthcare.

## 7. Future Perspective

The future perspective of agarwood as an antimicrobial and anti-inflammatory agent is promising, based on findings driven by ongoing research. In addition to its possible potential applications in various fields, agarwood could potentially be used in pharmaceuticals, cosmetics, and food preservation due to its antimicrobial effects. It can also potentially be used as an alternative to synthetic agents. With increasing concerns over antibiotic resistance, natural products like agarwood could serve as alternatives to synthetic antimicrobial agents. Agarwood has also been traditionally used in various cultures for its anti-inflammatory properties; hence, it has the potential to treat inflammatory conditions such as arthritis, skin inflammation, and gastrointestinal disorders.

## Figures and Tables

**Figure 1 antibiotics-13-01074-f001:**
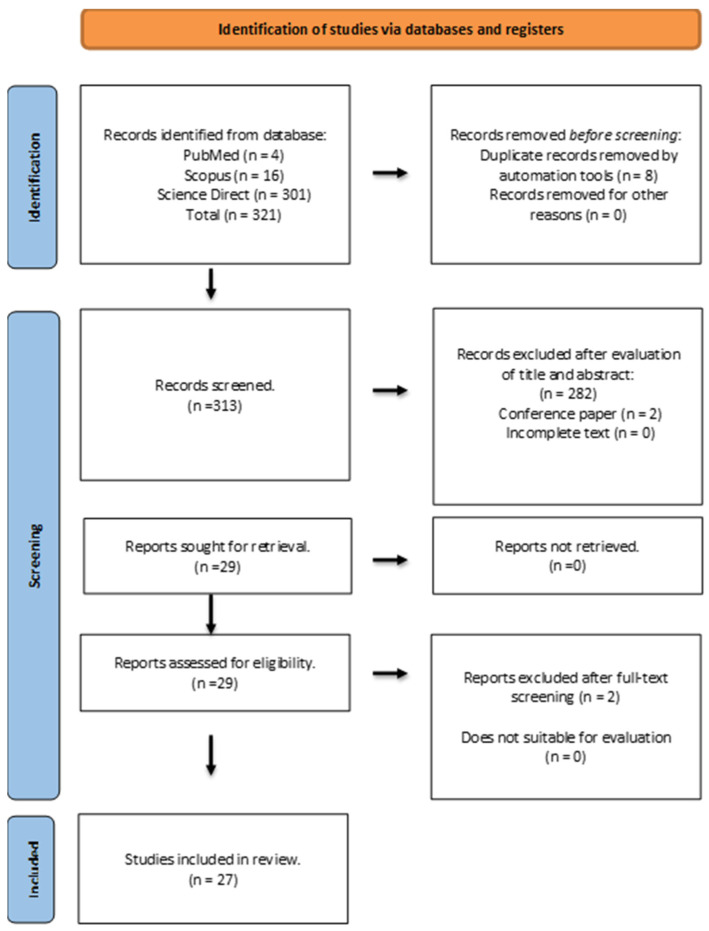
Flowchart of the search strategy.

**Table 1 antibiotics-13-01074-t001:** Tabulated summary of articles included in this review.

	Reference	Objective	Method	Findings	Conclusion/Recommendation
1	[13]	To investigate the chemical composition of volatile components and alcohol extracts from different agarwoods. To investigate the role of chromone compound 2-phenylethyl-benzopyran and the mechanism of agarwood formation.	GC-MS. Antimicrobial properties.	Volatile components, i.e., docosanoic acid 1,2,3-propanetriyl ester, oleic acid 3-(octadecyloxy) propyl ester, and 3-ethyl-5-(2-ethylbutyl)-octadecane. Alcohol extracts, i.e., benzoic acid ethyl ester, hexadecanoic acid ethyl ester, oleic acid, and n-hexadecanoic acid. The main active ingredients, i.e., sesquiterpenoids, aromatic species, and chromone compounds. The essential oil inhibited Gram-positive bacteria better than Gram-negative bacteria.	The study identified the chemical compositions of the best quality agarwood from different origins.
2	[14]	To identify the in vitro antimicrobial activity of ethanol extract from *A. agallocha* roots.	Disk diffusion method.	The ethanol extracts exhibited antimicrobial action against most of the tested microorganisms. Volatile components, i.e., docosanoic acid 1,2,3-propanetriyl ester, oleic acid 3-(octadecyloxy) propyl ester, and 3-ethyl-5-(2-ethylbutyl)-octadecane. Ethanol nextracts, i.e., benzoic acid ethyl ester, hexadecanoic acid ethyl ester, oleic acid, and n-hexadecanoic acid. Active ingredients, i.e., chromone compounds sesquiterpenoids, and aromatic species. The essential oil better inhibited Gram-positive than Gram-negative bacteria.	*A. agallocha* roots have a medicinal use.
3	[15]	To determine the content type of secondary metabolites and the antibacterial activity against *S. aureus* and *Proteus mirabilis* using the Kirby–Bauer disc diffusion method.	Phytochemical screening. Thin-layer chromatography. Kirby–Bauer disc diffusion.	The largest diameter zone was observed in *S. aureus*, at 15.80 mm, with *P. mirabilis* displaying a diamter of 15.19 mm.	Exthanolic extracts of Agarwood leaves exhibited activity against Gram-positive and Gram-negative bacteria.
4	[16]	To determine bioactive compounds of agarwood (*A. malaccensis*) ethanol extract and its antibacterial and antifungal activities against bacteria (*Staphylococcus epidermidis* ATCC 12228, *S. aureus* ATCC 25923, and *Propionibacterium acnes* ATCC 6919)/fungi (*C. albicans* ATCC 10231 and *Trichophyton* sp. ATCC 18748) species that commonly cause skin infection.	Ethanol extracts of agarwood leaves. Disc diffusion method (Kirby–Bauer test). Antifungal activity.	The maximum inhibitory zone toward *S. epidermis* and *P. acnes* was at 20% concentration. The inhibitory zone of *C. albicans* was classified as intermediate, at 20% concentration. *Trichopyiton* sp. was classified as resistant at all concentrations used. *S. aureus* was classified as susceptible at 5% concentration and intermediate at 2.5% concentration. Agarwood leaf ethanol extracts produced nine biologically active compounds.	The ethanol extract from *A. malaccensis* exhibited antibacterial and antifungal activities. The chemical compounds were flavonoids, tannins, and triterpenoids.
5	[17]	To examine antibacterial activity of the *A. crassna* leaf extract against *S. epidermidis* and its underlying mechanism. The antioxidant activity and acute toxicity were studied as well.	FRAP, ABTS, and DPPH scavenging methods. Disc diffusion assay and the minimum inhibitory concentration (MIC).	*S. epidermidis* was susceptible, with an MIC and MBC of 6 and 12 mg/mL, respesctively. The *S. epidermidis* cell wall ruptured after 24 h of treatment. No sign of toxicity or death was observed at the doses of 2000 and 15,000 mg/kg body weight in mice.	The aqueous extract of *A. crassna* leaves exhibits antimicrobial action against *S. epidermidis* in vitro.
6	[18]	To evaluate the combination effects of *A. malaccensis* extract with polymyxins against *Acinetobacter baumannii* and *Klebsiella pneumoniae*.	In vitro time-kill studies. GC-MS analysis.	The crude extract alone or in combination with polymyxin B inhibited the bacteria growth over 24 h. Over 60 constituents were determined, with phytol and 9,12-octadecadienal as the major components.	Crude extract alone or combined with polymyxin B enhanced bacterial killing compared to polymyxin B alone. Phytol, 9,12-octadecadienal, oleic acid, n-decanoic acid, n-hexadecanoic acid, and squalene likely contributed to the antimicrobial activity. *A. malaccensis* leaf extract is a promising candidate as an antimicrobial agent, particularly against *A. baumannii and K. pneumoniae*.
7	[19]	To evaluate the potential of using *A. malaccensis* leaf extract as a biogenic medium to generate CuO NPs with antimicrobial potential.	The addition of 5 mM copper sulfate (CuSO_4_.5H_2_O) as the precursor to *A. malaccensis* leaf extract to study the generation of CuO NPs under different incubation conditions, e.g., crude extract preparation, precursor concentration, and incubation temperature.	A total of 5 m of M CuSO_4_.5H_2_O reacted with the boiled leaf extract at pH 6, resulting in a high formation of CuO NPs. The nanoparticles are primarily spherical. Amounts of 20 μL and 40 μL of 70 μg/μL CuO NPs demonstrated strong suppression of *B. subtilis*.	*A. malaccensis* leaf extract was used as a reducing agent to create CuO NPs in a simple, economical, and sustainable manner. To ascertain these nanoparticles’ antibacterial effectiveness against a wider variety of microbial infections, more research is required.
8	[20]	To report the chemical constituents and antimicrobial activity of essential oil hydrodistilled from the leaves and trunk of *A. banaensis* P.H.Hô (*Thymelaeceae*) from Vietnam.	Hydrodistillation of the essential oils. Three Gram-positive bacteria, three Gram-negative bacteria, and yeast were used in this study. Agar well diffusion and broth microdilution methods.	Trunk essential oil displays antimicrobial action against *S. aureus*, with an IC50 value of 153.7 μg/mL and a MIC value of around 256.0 μg/mL. The leaf essential oil did not inhibit the tested microorganisms.	The current outcome creates a possibility for further exploitation of the trunk essential oil of *A. banaensis*.
9	[21]	To determine the potential of oleanane triterpenoids (1-oxo-β-amyrin, hederagenin-an, 3β-acetoxyfriedelane, and ursolic acid) from agarwood as a COVID-19 antiviral by in silico study.	Molecular docking, prediction of Lipinski rules of five, and prediction of ADME. Main protease (Mpro) COVID-19 was used as a receptor	The tested ligand has a higher affinity for the major protease (Mpro) receptor than lopinavir or remdesivir.	Lipinski’s rule of five indicates that hederagenin-an has the potential to be developed as an oral COVID-19 antiviral medication.
10	[22]	To investigate the secondary metabolite classes and antimicrobial activity of distilled water extracts of agarwood (*A. malaccensis* Lamk).	*A. malaccensis* Lamk leaves, stems, barkless stems, and bark. Hydrodistillation method and disc diffusion method (Kirby–Bauer test).	Agarwood exhibits antimicrobial properties that can combat *Streptococcus mutant* bacteria.	Bark, peeled stems, leaves, and trunks all contained glycosides.
11	[23]	To determine the anticancer, antioxidant, and antimicrobial properties of the sesquiterpene β-caryophyllene from *A. crassna*.	FT-IR, NMR, and MS. Antimicrobial study.	Selective antimicrobial effect of β-caryophyllene was observed against *S. aureus* (MIC 3 ± 1.0 µM), with a more pronounced anti-fungal activity than kanamycin. β-caryophyllene showed strong suppression of colon cancer cells’ clonogenicity, migration, invasion, and spheroid formation.	β-caryophyllene is the active component of A. crassna; as a promising chemotherapeutic drug against colorectal cancers, β-Caryophyllene has the potential for further investigation.
12	[24]	To determine the antibacterial activity of agarwood bouya (*Aquilaria agallocha*) oil nanoemulsion against multidrug-resistant bacteria (MDR) and antibiotic-nonresistant bacteria.	Disk diffusion method.	The 20% agarwood bouya oil nanoemulsion could only suppress *E. coli* ATCC 43300, which produces ESBLs, by 3.3 mm, while *S. aureus* ATCC 25923 and *K. pneumoniae* ATCC 8724 have inhibition zones of 2.6 mm and 3.3 mm, respectively. The non-resistant *E. coli* ATCC 25922 bacteria showed active inhibition, with an inhibition zone of 13.3 mm.	The oil nanoemulsion’s ability to inhibit multidrug-resistant bacteria (MDR) will increase with its concentration.
13	[25]	To confirm the silver nanoparticles (AgNPs) formation and their biophysical characterization. To evaluate the larvicidal and pupicidal toxicity of *A. sinensis* essential oil (AsEO), *P. cablin* essential oil (PcEO), and biosynthesized AgNPs against larvae and pupae of the dengue and zika virus vector *Aedes albopictus*.	Synthesis and confirmation of AgNPs formation. Larvicidal and pupicidal bioassays. Histological analysis.	Biofabricated AgNPs showed the highest toxicity against *Ae. albopictus* larvae and pupae. Compared to the essential oils (AsEO and PcEO), the artificial AgNPs had a greater impact on the brush boundary and epithelial cells.	AgNPs made from the essential oils of *A. sinensis* and *P. cablin* may be used as a biopesticide to control mosquitoes in a more economical and safe manner. To fully understand how Ag nanoparticles work against mosquito vectors, more research is required.
14	[26]	To test the quality of the agarwood originated from *A. sinensis*, stimulated by the chemical method (S1), compared with that of wild agarwood (S2), using healthy trees (S3) as controls. To determine antimicrobial activities of agarwood essential oils originating from *A. sinensis*.	GC-MS. Agar well diffusion method. MIC and minimum bactericidal concentration (MBC) assay.	The essential oil is rich in sesquiterpenes and aromatic constituents, fatty acids, and alkanes. S1 and S2 essential oils displayed better inhibition activities against *B. subtilis* and *S. aureus*.	The essential oil originated from *A. sinensis* stimulated by the chemical method shows high similarity to wild agarwood. To find the right chemical agents and timeframe for improving agarwood development, more research is needed.
15	[27]	To characterize and compare the composition and antimicrobial activity of essential oils obtained from agarwood originating from *A. sinensis* (Lour.) Gilg, induced by a biological agent of agarwood, *Lasiodiplodia theobromae* (F), to those from wild agarwood (W) and uninoculated healthy trees (H).	GC-MS. Agar well diffusion method. Antifungal activity using MIC and MFC values.	F’s essential oil had a similar composition to that of W, rich in sesquiterpenes and aromatic constituents. H’s essential oil was abundant in alkanes. F and W were more effective at inhibiting *L. theobromae*, *F. oxysporum*, and *C. albicans*. *L. theobromae* displayed a high similarity to W, both in chemical composition and in antimicrobial activity.	The fungus-induced agarwood technique may be used to produce both agarwood and essential oils from Aquilaria trees.
16	[28]	To analyze the effectiveness of *G. versteegii* fruit extract as an antibacterial agent against *E. coli* and *S. aureus* and to identify the chemical compounds of the fruit.	n-hexane, dichloromethane and methanol extraction. GC-MS.	The dichloromethane extract at 40% exhibited the strongest antimicrobial activity against *S. aureus* compared to that against *E. coli*. *G. versteegii* fruit demonstrated both complete and partial inhibition against *S. aureus* and *E. coli*, respectively. Palmitic, oleic, and stearic acid, bis-(2-ethylhexyl) phthalate, 2,3-dihydro-3,5dihydroxy-6-methyl-4H-pyran-4-one, methyl octadec-9-enoate, squalene, and 2monopalmitin derivates were identified.	
17	[29]	To isolate, identify, and report on the biological activities of four new chromone derivatives and seven known analogues.	Spectroscopic methods. Anti-inflammatory activities.	Four novel 5,6,7,8-tetrahydro-2-(2-phenylethyl) chromones were recovered. Two chromone monomeric units are joined by a (5,5″)-carbon–carbon bond in Compound 1. Compound 2 exhibited notable anti-inflammatory properties, with an IC50 value of 3.46 μM.	From the agarwood made with Agar-Wit of *A. sinensis*, eleven 2-(2-phenylethyl)chromones (1–11), including four that have not yet been reported, were separated and identified. By preventing LPS-induced NO release in RAW 264.7 cells, all of the novel compounds demonstrated notable anti-inflammatory properties.
18	[30]	To investigate the chemical constituents and anti-inflammatory agents of agarwood produced via the whole-tree agarwood-inducing technique (Agar-Wit) from *A. sinensis*.	Column chromatographic technique and semi-preparation HPLC. Anti-inflammatory activity of LPS-induced RAW264.7 cells.	Eleven sesquiterpenes were identified. Baimuxinol (1) was identified as a new natural product. Compounds [petafolia A (4), (4αβ,7β, 8αβ)-3,4,4α,5,6,7,8,8α-octahydro-7-[1-(hydroxymethyl) ethenyl]-4α-methylnaphthalene-1-carboxaldehyde (9) and 12-hydroxy-4(5),11(13)-eudesmadien-15-al(10) were found. Compound 1,4 and 9 displayed anti-inflammatory activities.	After being isolated and identified, eleven sesquiterpenes demonstrated notable anti-inflammatory properties by preventing RAW 264.7 cells from releasing NO in response to LPS.
19	[31]	To isolate a flavonoid compound, pilloin, from *A. sinensis* and investigate its anti-inflammatory activity in bacterial LPS-induced RAW 264.7 macrophages and septic mice.	Anti-inflammatory activity in bacterial LPS-induced RAW 264.7 macrophages and septic mice.	In LPS-activated macrophages, pilloin blocked the NF-κB and MAPK signaling pathways. TNF-α, IL-6, iNOS and COX-2 were downregulated by pilloin. Pilloin inhibited the phenotypes and activities of activated macrophages. Via in vivo study, pilloin was shown to reduce the amount of cytokines TNF-α and IL-6.	Pilloin is a potential anti-inflammatory compound.
20	[32]	To isolate and structurally elucidate the compounds, as well as their inhibitory effects, on NO production in LPS-stimulated RAW 264.7 cells.	Ethanol extraction. Cell culture, viability assay, and measurement of NO production. LCMS.	Fifteen undescribed 2-(2-phenylethyl)chromone dimers and two known analogues were isolated. Significant inhibition of NO production in LPS-stimulated RAW 264.7 cells by the isolated compounds.	Significant NO generation suppression was demonstrated by 2-(2-phenylethyl)chromone dimers, suggesting that they may be used as a precursor to anti-inflammatory drugs.
21	[33]	To structuraly elucidate the compounds and the inhibitory activities of all isolates from *A. sinensis*.	Spectroscopic and MS analyses. Inhibitory activities of LPS-induced NF-κB activation of macrophages.	7-methoxy-2-(2-phenylethyl)-chromone, 6,7-dimethoxy-2-[2-(40-methoxyphenyl)ethyl]chromone, and 6,7-dimethoxy-2-(2-phenylethyl)chromone prevented LPS-stimulated RAW 264.7 macrophages from activating NF-κB. 7-methoxy-2-(2-phenylethyl)chromone, 7-dimethoxy-2-(2- phenylethyl)chromone 5,6-dihydroxy-2-[2-(30 -hydroxy 4 0 -methoxyphenyl)ethyl]chromone, and 6,7-dimethoxy-2-[2-(40-methoxyphenyl)ethyl]chromone did not cause cytotoxicity against RAW 264.7 cells and was able to inhibit LPS-induced NO generation in these cells after 24 h.	Agarwood and its derivatives merit additional biological research and may be developed as viable treatments or preventative measures for a number of inflammatory illnesses. There is little doubt that more research should be conducted on the structure–activity relationship (SAR) of these isolated compounds in terms of their anti-inflammatory potential.
22	[34]	To isolate 16 new 2-(2-phenylethyl)chromone dimers, including four pairs of enantiomers (1a/1b, 3a/3b, 6a/6b, and 8a/8b), along with eight optically pure analogues (2, 4, 5, 7, and 9–12), from *A. sinensis*.	Spectroscopic analysis.	Compounds 11 and 12 have an exceptional 6,7-dihydro-5H-1,4-dioxepine moiety in their structures. Compounds 1–10 have a unique 3,4-dihydro-2H-pyran ring linkage joining two 2-(2-phenylethyl)chromone monomeric units. Compounds 1a/1b, 2, 3a/3b, 5, 7, 8a/8b, and 10–12 significantly inhibited the generation of NO.	
23	[35]	To report the isolation, structural elucidation, and anti-inflammatory activity of these compounds.	Methanol extraction. Spectroscopic analyses. Measurement of O_2_ generation.	One compound demonstrated an over 80% reduction of human neutrophils’ production of superoxide anion.	Further research into *A. sinensis’s* resinous wood and its constituents—particularly 8 and 18—may be warranted to confirm their use as viable options for the management or avoidance of a number of inflammatory illnesses.
24	[36]	To describe the structural elucidation of the compounds, numbered 1 through 3, and the inhibitory activities of all isolates regarding superoxide generation and elastase release via neutrophils.	Methanol extraction. Biological assay. NMR and MS.	Human neutrophils’ production of superoxide anion was inhibited by compounds 2, 3, 5, 6, and 8–10. Compounds 3, 6, 8, 10, and 19 prevented the release of elastase generated by fMLP/CB.	Bioactive isolates, particularly 2, 3, 5, 6, 8, 9, 10, and 19, have the potential to be further developed as potential treatments or preventative measures for a variety of inflammatory illnesses.
25	[37]	To isolate and elucidate the structure of five new 2-(2-phenylethyl)chromones (1−5) and to describe their inhibitory effects on nitric oxide (NO) production in LPS-stimulated RAW 264.7 cells.	NMR, UV, IR, and MS analyses. Electronic circular dichroism (ECD) calculations. Cell viability assay. Griess assay.	Eleven recognised compounds and five novel 2-(2-phenylethyl)chromone derivatives were identified from Chinese agarwood. Compounds 2–4, 11, 12, and 15 demonstrated a considerable reduction of NO generation in LPS-stimulated RAW 264.7 cells.	The anti-inflammatory properties of 5,6,7,8-tetrahydro-2-(2-phenylethyl)-chromones may be linked to the chlorine substituent and epoxy group on the A-ring.
26	[38]	To investigate the antipyretic, analgesic, anti-inflammatory, and anti-oxidative properties of the extract of *A. crassna* leaves with a wide dose range in rodents.	In vivo study using aqueous extract of A. crassna leaves (ACE). Antipyretic assay, analgesic and anti-inflammatory activities. DPPH assay.	ACE significantly reduced the rectal temperature, Mice were less sensitive to heat. No anti-inflammatory activity was observed. Anti-oxidative activity was observed for ACE, with an IC50 value of 47.18 μg/mL. No behavioral changed. Significant body weight loss.	No toxicity was identified. Extracts from *A. crassna* leaves have analgesic, antipyretic, and antioxidative qualities, but no anti-inflammatory effects.
27	[39]	To analyze the chemical constituents of incense smoke generated by the whole-tree agarwood-inducing technique (AWIT), agarwood induced by axe wounds (AAW), burning-chisel drilling agarwood (BCDA), and the wood of *A. sinensis* trees (AS), To investigate the effect of the chemical constituents on TNF-α and IL-1α release in LPS-stimulated RAW 264.7 cells.	Chromatographic separation. TNF-α and IL-1α production.	A total of 484 compounds were identified. A total of 61 aromatic compounds were identified. AAW, AWIT, and indomethacin have anti-inflammatory properties.	By preventing the release of TNF-α and IL-1α in RAW 264.7 cells produced by LPS, agarwood incense smoke demonstrated anti-inflammatory properties.

## Data Availability

Not applicable.

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
