# Peer review of "The Therapeutic Potential of Agarwood as an Antimicrobial and Anti-Inflammatory Agent: A Scoping Review"

_antibiotics, 2024, doi:10.3390/antibiotics13111074_

Round 1
Reviewer 1 Report
Comments and Suggestions for Authors
The work entitled “The Therapeutic Potential of Agarwood as an Antimicrobial and Anti-inflammatory Agent: A Scoping Review” reports on the potential health benefits of agarwood as an antimicrobial and anti-inflammatory agent, focusing on research conducted between 2013 and 2023 and its applications in the biomedical field. The work is very well organized and the subject is of interest. The paper is also very well written and clear. However, prior to publication I recommend minor revisions as follows:
- Reduce abstract. There is too much information being given, without need. It is possible to summarize the subject in fewer sentences and make it more attractive to the readers.
- The methodology employed is well thought out. However, there is no clear indication why were some articles rejected and others not.
- The discussion is very well done. But the presentation of the results would benefit from images and schematic representations of the main findings.
- Future perspectives are missing and should be included in order to understand if the use of agarwood has a real impact in the investigation and if it has a place to continue being used.
Author Response
Comments 1: Reduce abstract. There is too much information being given, without need. It is possible to summarize the subject in fewer sentences and make it more attractive to the readers.
Response 1: Thank you for pointing this out. We have reduced the abstract accordingly to make the information provided more precise and attractive to the reader.
Comments 2: The methodology employed is well thought out. However, there is no clear indication why were some articles rejected and others not.
Response 2: We are aware of what the reviewer voiced regarding the indication of why some articles were rejected and others not. However, there are several reasons why we rejected several articles during the review process for this review paper based on several criteria, which include but are not limited to the following:
a. The relevancy of the research article. The article may not be relevant to the scope or focus of the review paper. It might discuss a topic that is tangential or unrelated to the main theme of the review.
b. Clarity and structure: Poorly written articles that are difficult to understand or lack coherent structure may be rejected. This includes issues with grammar, readability, and organization.
c. Editorial Decision: Ultimately, the decision to reject an article for a review paper can also depend on the editorial judgment and goals of the journal or publication.
Comments 3: The discussion is very well done. But the presentation of the results would benefit from images and schematic representations of the main findings.
Response 3: We agree that including images and schematic representations makes the findings more interesting because they can be seen visually. However, for this review, we take the approach of providing information in text for several reasons:
a. Precision and Clarity: Textual descriptions often provide more precise and nuanced explanations of results than images or schematics. They can convey subtle differences and contextual information that might be missed in visual representations.
b. Integration with Text: Academic papers require results to be integrated seamlessly into the paper's narrative. Textual descriptions allow authors to discuss and interpret the results about existing literature, hypotheses, and implications.
c. Detail and Depth: Textual descriptions allow authors to provide detailed explanations, including nuances, limitations, and interpretations of the results. This depth of analysis is often crucial in academic writing to support the conclusions drawn from the data.
Although images and schematics can be useful to illustrate certain things or summarize complex data, not everything can be done for review papers that look at diverse studies from the aspect of species used, extraction, and methodology used by many researchers. Therefore, textual descriptions remain the preferred method for presenting results in such contexts, ensuring the information is comprehensible, detailed, and integrative within the scholarly discourse.
Comments 4: Future perspectives are missing and should be included in order to understand if the use of agarwood has a real impact in the investigation and if it has a place to continue being used.
Response 4: Thank you! We found your comments extremely helpful and have included the future perspectives of agarwood as subtitle no. 5. (Line 369-379)
Reviewer 2 Report
Comments and Suggestions for Authors
The article discusses the potential health benefits of agarwood (Aquilaria spp.) as an antimicrobial and anti-inflammatory agent. It reviews original research articles from 2013 to 2023, focusing on the effectiveness of agarwood against various microbes and its anti-inflammatory properties. Agarwood has shown promise as a biopesticide and as an inhibitor of fungi and bacteria, including Bacillus subtilis and Staphylococcus aureus. The review concludes that, despite the absence of clinical trials, agarwood exhibits significant antimicrobial and anti-inflammatory properties, meriting further biomedical research. The article also highlights some weaknesses:
-
The review only includes 27 full-text articles, which may not provide a comprehensive overview of agarwood's potential. A larger dataset could yield more robust conclusions.
-
The names of the plants should be written in full, including the genus and binomial nomenclature, when first mentioned. The accepted names of the species should be verified using https://www.worldfloraonline.org/. For example, Aquilaria sinensis Merr. (Thymelaeaceae). All agarwood species from the articles should be checked to determine if they are synonyms of Aquilaria species.
-
The methodologies and quality of the included studies vary, which can affect the reliability of the review’s conclusions. Some studies may have methodological flaws or biases that were not adequately addressed. For example, when summarizing antimicrobial activity, ensure that all cited papers follow the proposed/suggested effective concentration for antimicrobial activity, such as considering an extract active if the MIC concentration is < 100 µg/ml. Similarly, for the disk diffusion method, the antimicrobial effectiveness should be carefully reviewed based on the zone of inhibition. Concerns also arise when the tested materials vary (e.g., extract, essential oil, nanoemulsion). The activities of these materials need to be justified.
-
Authors should separate information on antimicrobial and anti-inflammatory activities. The same issue was observed for anti-inflammatory testing. The basis for selecting articles that claim agarwood has good anti-inflammatory activities should be clearly stated.
Author Response
Comments 1: The review only includes 27 full-text articles, which may not provide a comprehensive overview of agarwood's potential. A larger dataset could yield more robust conclusions.
Response 1: We understand the concern raised by the reviewer for only selecting 27 articles, but this selection has been made according to the consensus of all the authors involved. We have examined 321 publications in the initial stage before screening. We also limit publication for a period of 10 years for several reasons:
a. Relevance and Currency: Including recent publications ensures that the review reflects the most current state of research and knowledge in that particular area. This selectivity is essential in rapidly advancing fields where older publications may not represent the latest advancements or perspectives.
b. Focus and Scope: Limiting the timeframe helps maintain focus on recent developments and trends within a manageable scope. This approach allows authors to delve deeper into the most pertinent and up-to-date research without being overwhelmed by the volume of older literature.
c. Quality and Methodological Standards: Recent publications are more likely to adhere to current research methodological standards and quality controls. This standard ensures that the review includes studies that meet rigorous criteria for reliability and validity.
d. Reader Expectations: Readers of review papers typically expect a synthesis of the latest findings and trends. Limiting the review to recent publications meets this expectation by providing insights into current debates, controversies, and emerging concepts.
e. Practicality and Manageability: Reviewing a limited timeframe (e.g., the past ten years) makes the task more feasible in terms of time and effort. It allows authors to conduct a thorough analysis within a reasonable timeframe, considering the volume of literature published annually.
While the ten-year limit is standard, it can vary depending on the field, topic, and specific requirements of the review paper. Ultimately, the timeframe chosen should balance comprehensiveness with relevance to provide the most insightful and valuable synthesis of the literature.
Comments 2: The names of the plants should be written in full, including the genus and binomial nomenclature, when first mentioned. The accepted names of the species should be verified using https://www.worldfloraonline.org/. For example, Aquilaria sinensis Merr. (Thymelaeaceae). All agarwood species from the articles should be checked to determine if they are synonyms of Aquilaria species.
Response 2: Thank you for the valuable suggestions. We have checked individually for each published article to ensure that the species used is based on the publication made by the original author of the study. However, some researchers only place "spp." without giving the specific name of the species. To avoid confusion for readers, we ensure that no changes are made and that this scoping review is based on the original articles' findings. Ultimately, we do our best to ensure that genus and binomial nomenclature for this scoping review is guided by journal-specific requirements and balanced with considerations of clarity, accuracy, and readability for the intended audience.
Comments 3: The methodologies and quality of the included studies vary, which can affect the reliability of the review’s conclusions. Some studies may have methodological flaws or biases that were not adequately addressed. For example, when summarizing antimicrobial activity, ensure that all cited papers follow the proposed/suggested effective concentration for antimicrobial activity, such as considering an extract active if the MIC concentration is < 100 µg/ml. Similarly, for the disk diffusion method, the antimicrobial effectiveness should be carefully reviewed based on the zone of inhibition. Concerns also arise when the tested materials vary (e.g., extract, essential oil, nanoemulsion). The activities of these materials need to be justified.
Response 3: We understand what concerns the reviewer, but since this manuscript is a scoping review and not a systematic review that focuses more on the similarity of the value of the data obtained, we are only able to make a comparison based on the data provided by the original researcher. This manuscript cannot be written as a systematic review because the extraction method, species, and research methodology between one researcher and another are not uniform. Scoping reviews often address broader research questions and may include a wide range of study designs and types of literature that differ from systematic reviews. Systematic review goals are to provide a comprehensive summary of evidence on a narrowly defined topic using predefined criteria and methods to minimize bias. While scoping and systematic reviews are methods of synthesizing research literature, their purpose, focus, inclusion criteria, search strategies, synthesis methods, and intended outputs differ.
Comments 4: Authors should separate information on antimicrobial and anti-inflammatory activities. The same issue was observed for anti-inflammatory testing. The basis for selecting articles that claim agarwood has good anti-inflammatory activities should be clearly stated.
Response 4: At the initial stage, we tried to separate antimicrobial and anti-inflammatory activities. However, during the write-up process, we found that combining antimicrobial and anti-inflammatory activities offers several advantages and synergistic effects. It is suitable to combine them for the following reasons:
a. Comprehensive Treatment Approach: Many infections are accompanied by inflammation, and vice versa. By targeting both antimicrobial activities to combat the infection and anti-inflammatory activity to reduce inflammation and associated symptoms, a treatment can provide a morecomprehensive therapeutic approach.
b. Reduced Complications: Inflammation is a natural response of the immune system to infection, but excessive or prolonged inflammation can lead to tissue damage and complications. Anti-inflammatory agents can help mitigate these effects, potentially reducing the severity and duration of symptoms.
c. Enhanced Efficacy: Some compounds exhibit both antimicrobial and anti-inflammatory properties synergistically. This dual action can enhance the overall efficacy of the treatment compared to using separate agents for each purpose.
d. Potential for Novel Therapeutics: Research into compounds or formulations that exhibit both antimicrobial and anti-inflammatory properties can lead to the development of novel therapeutic agents with unique mechanisms of action and improved therapeutic profiles.
Overall, combining antimicrobial and anti-inflammatory activities represents a promising medical and pharmaceutical research approach, aiming to provide more effective, targeted, and holistic treatments for infectious and inflammatory diseases.
Reviewer 3 Report
Comments and Suggestions for Authors
This manuscript reviews the potential health benefits of agarwood as an antimicrobial and anti-inflammatory agent. The authors conclude that agarwood and its isolates are worthy of further biomedical investigation and could be developed as potential candidates for the treatment or prevention of various microbial and inflammatory diseases.
A few concerns for the authors.
1. Please check grammatical and spelling errors. For example: 1) Page 2, line 52, “have been taught to have antimicrobial properties.”; 2) Page 2, line 55-56, “allowing internal organelles to seep out and eventually die.” I guess here should be the cells eventually die but it does not look like this from the sentence. 3) Page 2, line 56, “interference with microbial enzymes where the components of agarwood…” 4) Page 2, line 60, “by compounds found in agarwood may affect the pathogenicity”, should be “that may affect…”? There are some other similar mistakes throughout the manuscript, and I would suggest the authors read it through carefully and correct those mistakes.
2. Page 3, Figure 1, in the screening process, the authors excluded most of the articles that they found (282 out of 313). What are the reasons for excluding them? More explanations will be needed.
3. Page 4-13, Table 1, there are several issues. 1) There is a column called “title” but only the reference numbers are there. It would be nice to have the real titles shown up in this column. There is enough space for each title anyways. 2) The reference number should be in a separate column. 3) In the column of findings, everything needs to be more concise with the more focused antimicrobial and anti-inflammatory activities of Agarwood. 4) It would be very helpful to list the bioactive compounds or extracts, and the possible mechanisms of action.
Comments on the Quality of English Language1. Please check grammatical and spelling errors. For example: 1) Page 2, line 52, “have been taught to have antimicrobial properties.”; 2) Page 2, line 55-56, “allowing internal organelles to seep out and eventually die.” I guess here should be the cells eventually die but it does not look like this from the sentence. 3) Page 2, line 56, “interference with microbial enzymes where the components of agarwood…” 4) Page 2, line 60, “by compounds found in agarwood may affect the pathogenicity”, should be “that may affect…”? There are some other similar mistakes throughout the manuscript, and I would suggest the authors read it through carefully and correct those mistakes.
Author Response
Comment 1: Please check grammatical and spelling errors. For example: 1) Page 2, line 52, “have been taught to have antimicrobial properties.”; 2) Page 2, line 55-56, “allowing internal organelles to seep out and eventually die.” I guess here should be the cells eventually die but it does not look like this from the sentence. 3) Page 2, line 56, “interference with microbial enzymes where the components of agarwood…” 4) Page 2, line 60, “by compounds found in agarwood may affect the pathogenicity”, should be “that may affect…”? There are some other similar mistakes throughout the manuscript, and I would suggest the authors read it through carefully and correct those mistakes.
Response 1: Thank you for raising the matter. We have repaired the entire manuscript, preferably in terms of grammar and spelling errors.
Comment 2: Page 3, Figure 1, in the screening process, the authors excluded most of the articles that they found (282 out of 313). What are the reasons for excluding them? More explanations will be needed.
Response 2: Thank you for the comments. To clarify our explanation, we have included the reason why we excluded many articles that were screened in the initial stage in the "Results" section (lines 100 -131).
Comment 3: Page 4-13, Table 1, there are several issues. 1) There is a column called “title” but only the reference numbers are there. It would be nice to have the real titles shown up in this column. There is enough space for each title anyways. 2) The reference number should be in a separate column. 3) In the column of findings, everything needs to be more concise with the more focused antimicrobial and anti-inflammatory activities of Agarwood. 4) It would be very helpful to list the bioactive compounds or extracts, and the possible mechanisms of action.
Response 3: Thank you for your suggestions. At the initial stage, we wanted to include the article's title in the column, but since the review paper that had been published before only put the "Reference" as a column heading, we removed the title from the column according to the publication. We have also changed the column heading to "Reference."
Thank you for the comment to refine the review of this article. We have taken the step of purification by only including the necessary elements for this write-up, which are related to the antimicrobial and anti-inflammatory effects as well as the compound extract mentioned in the selected research article.
Round 2
Reviewer 2 Report
Comments and Suggestions for Authors
Even it is not a systematic review, careful review manuscript in order to give better understanding of the bioactivity of the agarwood must be clearly defined comprehensively, which authors did not do in the revised version.
Author Response
Comment 1: Even it is not a systematic review, careful review manuscript in order to give better understanding of the bioactivity of the agarwood must be clearly defined comprehensively, which authors did not do in the revised version.
Response 1: Thank you for pointing this out. We have made the best changes in the manuscript.
We have included improvements in the manuscript in the column "Discussion related to why the combination of antimicrobial and anti-inflammatory is done together (lines 140-151). In addition, additional statements related to disc diffusion assay were also included (lines 195-197). A little addition pertaining to the combination of the extract with polymyxin B was included (lines 204-206).
We have added the subtitle "Limitations of the Study" (lines 397-402) to this review paper. It explains the question raised as to why reaching more accurate conclusions is difficult.
In addition, we also included the reason why only 27 papers were selected for this paper review in the "Results" section (lines 88 – 111).
Reviewer 3 Report
Comments and Suggestions for Authors
All concerns addressed. No more questions.
Author Response
Comment 1: All concerns addressed. No more questions.
Response 1: Thank you for the feedback.
Round 3
Reviewer 2 Report
Comments and Suggestions for Authors
I have noted that the similarity percentage is 56%, which is considerably higher than the acceptable threshold. This raises concerns about potential overlap with previously published works.
Author Response
Comment 1: I have noted that the similarity percentage is 56%, which is considerably higher than the acceptable threshold. This raises concerns about potential overlap with previously published works.
Response 1: Thank you for pointing this out. We apologise for the relatively large number of similarities in this manuscript. We have rephrased it in its entirety without changing the meaning of the sentence to ensure that the level of similarity can be reduced to the lowest level after excluding affiliations, tables and references. We hope that you will be satisfied with the content of this manuscript.